# Determination of Vibration Picking Parameters of *Camellia oleifera* Fruit Based on Acceleration and Strain Response of Branches

**Delin Wu *** , **Enlong Zhao, Dong Fang, Shan Jiang, Cheng Wu, Weiwei Wang and Rongyan Wang**

School of Engineering, Anhui Agricultural University, Hefei 230036, China
* Correspondence: wudelin@ahau.edu.cn; Tel.: +86-158-0560-2399

**Abstract:** This study examines the means of reducing the damage to the branches of *Camellia oleifera* in the process of vibration picking and solving the problems of low equipment-development efficiency and slow product renewal caused by using traditional test methods to determine vibration picking parameters. In this study, the optimal vibration parameters were determined by using the self-response (branch acceleration and strain) law of the *Camellia oleifera* tree, and finite element analysis and experiments are used to solve this problem. The 3D model of *Camellia oleifera* was built by Solidworks. The natural frequencies of *Camellia oleifera* were analyzed by modal analysis, the vibration frequency and amplitude were determined by harmonic response analysis, and transient analysis was used to compare with the test results. The results show that the optimal vibration frequency range of *Camellia oleifera* is 4~10 Hz, and the average correlation coefficient between the maximum synthetic acceleration and the simulated value is 0.85, which shows that the model can reliably predict the vibration response. At the same time, the best vibration parameters were determined to be 9 Hz, 60 mm and 10 s. Under these parameters, the abscission rate of the *Camellia oleifera* fruit was 90%, and the damage rate of the flower bud was 13%. The mechanized picking effect of *Camellia oleifera* fruit was good. This study can quickly determine the vibration picking parameters of *Camellia oleifera* fruit and effectively improve the development speed of vibration picking of *Camellia oleifera* fruit.

**Keywords:** *Camellia oleifera*; modal analysis; harmonic response analysis; transient analysis; vibration parameters; mechanized picking



## 1. Introduction

*Camellia oleifera*, a unique high-quality edible oil plant in China, has important agricultural and socio-economic significance [1]. The oil content of *Camellia oleifera* seed is 25~35%, and the *Camellia oleifera* seed oil is considered to be the healthiest vegetable oil. The pressed *Camellia oleifera* seed oil mainly contains unsaturated fatty acids, such as oleic acid and linoleic acid [2,3].

In the process of the tea-oil industry development, the harvesting of the tea-oil fruits is the most expensive stage [4], which needs a lot of human, material and financial resources. Therefore, it is necessary to develop efficient harvesting methods to reduce the labor and economic costs, and solve the problem of labor shortage and improve production efficiency. In the past ten years, many mechanized fruit-picking machines have been developed and designed [5–8], among which the vibration type has been proved to be one of the most effective ways for mechanized fruit-picking [9–11]. The determination of optimal vibration parameters has been the focus of research.

Some researchers use the finite element method to simulate and optimize the harvesting system, and determine the impact of the harvesting parameters on the actual harvest by iterating a large number of designs in a shorter time [12–14]. The vibration harvesting of the Citrus crown was simulated by ANSYS, and the effects of the vibration frequency and penetration depth on fruit abscission were studied. The effects of the Ginkgo crown

structure on the vibration characteristics and the variation of damping with frequency were studied [15,16]. Combining a simulation with a dynamic model has obtained the best vibration frequency and amplitude of mechanized picking [17,18]. Although the finite element method is an effective method to study the vibration harvesting, the previous simulation research using the finite element method were based on various assumptions, which is different from the actual harvesting.

Therefore, some researchers focus on field experiments to study the effective harvesting methods [19–22]. Pu, Y measured the acceleration of the branches and fruits using accelerometers and found that the force of fruits inside the canopy was higher than the force of fruits at the edges [23,24]. The dynamic analysis of the Carya cathayensis was carried out by impact test [25], and the predicted optimal vibration frequency of the trunk vibrator was shown. The vibration transmission of a multi-directional trunk vibrator in the olive tree was analyzed [26], which showed that the vibration transmission was affected by the dynamic behavior of olive tree. In addition, Du, X.Q., et al. [27] compared the responses of walnut trees harvested with different eccentric weights. In these studies, the appropriate harvest parameters were obtained through experiments. However, the experimental method is complex, time-consuming, and difficult to implement, due to the influence of environment and time.

In previous studies, whether using the finite element method or the actual test method, the determination of the optimal working parameters is mostly based on the effect of the excitation device, and there is little research on the determination of the optimal working parameters from the tree response law. Considering the above problems, in this study: (1) A 3D model of tea oil tree was constructed; (2) The optimum vibration frequency and amplitude suitable for tea oil harvesting were determined through modal analysis and harmonic response analysis; (3) The corresponding relationship between simulation and actual test was compared, and the actual vibration response (acceleration, strain) of *Camellia oleifera* was predicted under vibration load; (4) Field experiments were conducted to verify the vibration frequency and amplitude, and to analyze the effect of the mechanized picking of *Camellia oleifera* fruits under different timings.

Compared with the previous relevant studies, the main contributions of this study are as follows:

1.  Proposing the method of combining a simulation with an experimental test to determine the best vibration parameters of *Camellia oleifera* mechanized picking;
2.  Considering the plastic deformation of the *Camellia oleifera* branches during vibration picking;
3.  Establishing the correspondence between the simulation and experimental results that can predict the actual vibration response of the tea oil trees.

## 2. Materials and Methods

### 2.1. Material

A 5-year-old tea oil tree in Tongcheng City, Anhui Province was obtained from Qinglongwan Agricultural Ecological Development Co., Ltd. (31.05° N, 116.95° E). The measuring instruments used included a vernier caliper with an accuracy of 0.01 mm, a tape measure with an accuracy of 1 mm and a plumb bob.

### 2.2. Establishment of Three-Dimensional Model of Camellia Oleifera

Firstly, the Cartesian coordinate system of the *Camellia oleifera* was determined, and the coordinate values of the characteristic points on the trunk and lateral branches of *Camellia oleifera* were measured with tape and plumb bob. Assuming that the trunk and branches of *Camellia oleifera* are round, we used a tape measure to measure the circumference of the trunk, so as to obtain the trunk diameter of *Camellia oleifera*. The diameter of the branches was measured with a vernier caliper. Then, the measurement points were drawn in the 3D sketch. By establishing a reference surface perpendicular to the 3D spline curve, the spline curve was used to link these feature point sketches. Finally, the 3D model of *Camellia*

*oleifera* was established by using the lofting function of SolidWorks through cross section and spline curve, as shown in Figure 1. The branches with a relatively small diameter and the front end of the branches were not modeled because their diameters were very small. In addition, the *Camellia oleifera* leaves and fruits were not modeled, because they have little impact on the simulation results, and their addition will cause a great burden on the simulation.

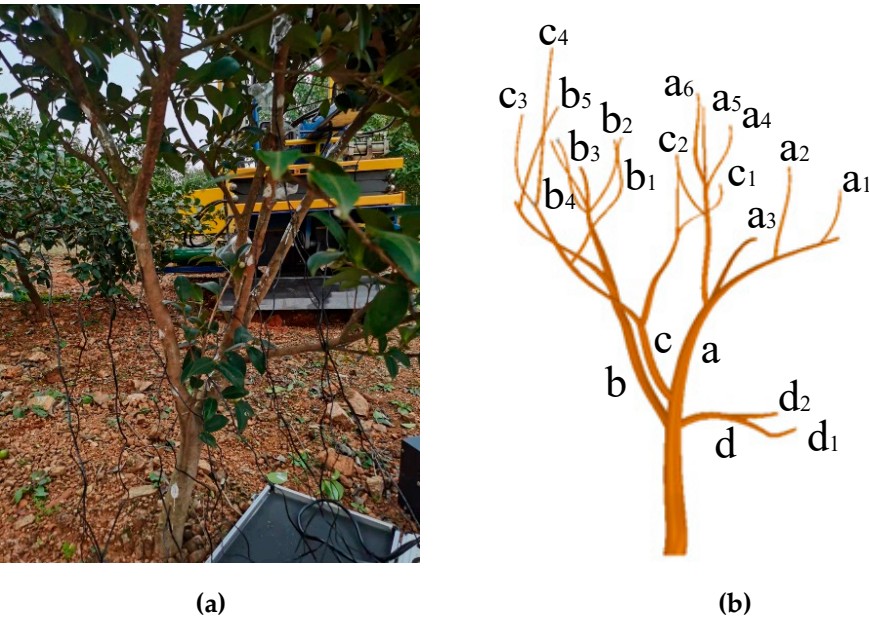

| **(a)** | **(b)** |

**Figure 1.** *Camellia oleifera* tree. (**a**) real photo; (**b**) 3D modeling. a, b, c, d represent different first-level branches of *Camellia oleifera* and $a_i$, $b_i$, $c_i$, $d_i$ represent different second-level branches of *Camellia oleifera*.

*2.3. Measurement of Physical Parameters of Camellia Oleifera*

Some of the physical properties of *Camellia oleifera*, such as elastic modulus, density, damping ratio and Poisson's ratio, were necessary for the simulation.

(1) The density of *Camellia oleifera* was measured by the immersion method. According to the water density of 1 g/cm$^3$, the value of the sample volume is equal to the value of the mass of water moving when the sample is immersed in water. The sample mass was also measured by an electronic balance with an accuracy of 0.001 g. Therefore, the density of *Camellia oleifera* (g/cm$^3$) was obtained by dividing the mass (g) by its volume (cm$^3$);

(2) The damping ratio of the *Camellia oleifera* is an inherent property of *Camellia oleifera*, which is measured by a vibration test. Decaying oscillation usually occurs when the damping system is subjected to an impact load, that is, when a sudden force is applied when an object suddenly accelerates or decelerates, or when the system is released from a displacement position relative to the equilibrium state in which its energy dissipates with time. The damping energy loss can be expressed as the damping ratio [28]. The relationship between the damping ratio of free vibration and logarithmic attenuation is shown in the Equations (1) and (2).

$$\delta = \frac{1}{N} \ln \frac{A_1}{A_{N+1}} \quad (N \geq 1) \tag{1}$$

$$\xi = \frac{\delta}{\sqrt{4\pi^2 + \delta^2}} \tag{2}$$

where $\delta$ is the logarithmic decay rate; $A_1$, $A_{N+1}$ are the first and $(N + 1)$ peak amplitudes of the vibration signal, respectively; $\xi$ is the damping ratio.

The damping ratio of each part of the branch is measured by the free attenuation of the branch through the rope-pull experiment. The acceleration sensor was arranged

in a suitable position on the sample tree, and the cotton rope was pulled to make the branch produce a certain displacement, and then the cotton rope was quickly loosened. The KDDASP instrument was used to collect the dynamic signal, and the experiment was repeated for three times on the sample tree. The group with a clear attenuation curve was selected for analysis;

(3) The elastic modulus was measured by the universal testing machine, as shown in Figure 2. According to the ASABE standard [29], the sample was loaded by two parallel plates on the machine at a loading rate of 10 mm/min. The failure force and associated deformation (in the vertical direction) were plotted as the output of the quasi-static tests, and then the elastic modulus was determined by using the Hertz theory [30];

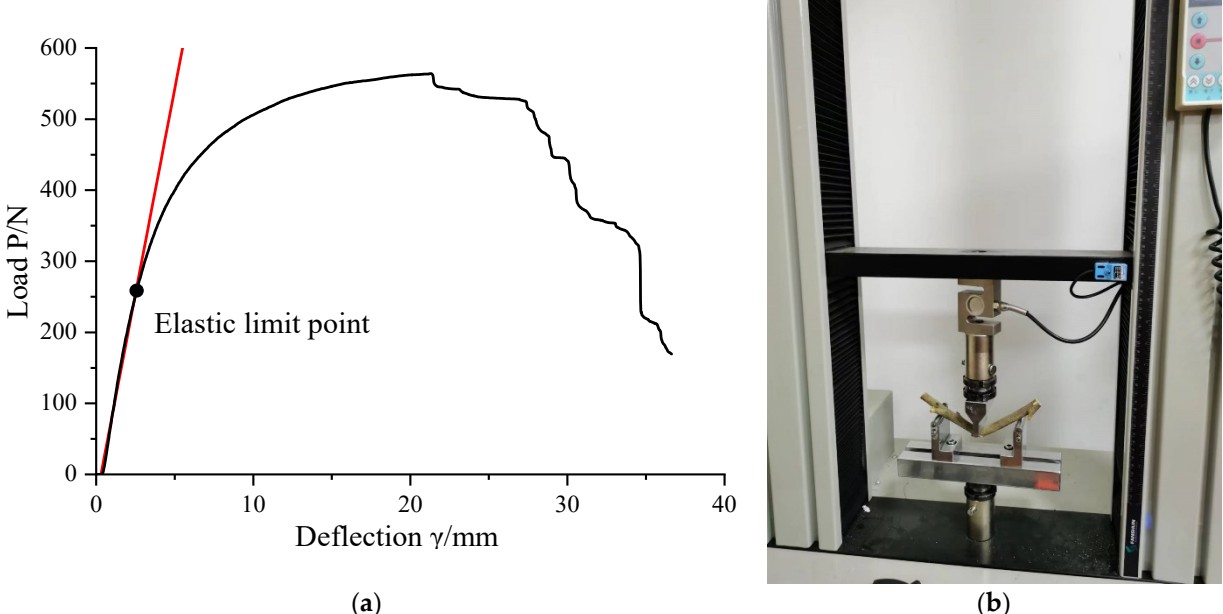

**(a)**

**(b)**

**Figure 2.** Three-point bending test. (**a**) Load displacement curve; (**b**) Test photo.

(4) The ultimate strain of the *Camellia oleifera* branch is defined as the strain at the critical point of plastic deformation of the branch, which is measured by the universal testing machine through the bending test. Record the vertical distance $h$ from the sample to the elastic limit point, the distance $L$ between the two fulcrums of the universal testing machine bending test fixture, and the radius of curvature $\rho$ of the sample being bent is calculated as follows:

$$\rho = \frac{12h}{L^2} \tag{3}$$

The strain is the deformation of the per unit length produced by the micro material elements in the beam structure after the beam structure is stressed, which is represented by $\varepsilon$ [31].

A micro segment is removed from the curved deformed branch, as shown in Figure 3, in which $O_1O_2$ is the neutral axis, and the length is $dx$. According to the basic assumptions of elasticity, the neutral axis length remains unchanged before and after the deformation of the micro segment, and the ab length at the bottom will change, and the amount of change is shown in (4):

$$\Delta s = L_{ab} - L_{o_1 o_2} = \left(\rho + \frac{D}{2}\right)d\theta - \rho d\theta = \frac{D}{2}d\theta \tag{4}$$

where $\Delta s$ is the change in length ab at the bottom; $\rho$ is the curvature radius of neutral layer after deformation; $D$ is the branch diameter:

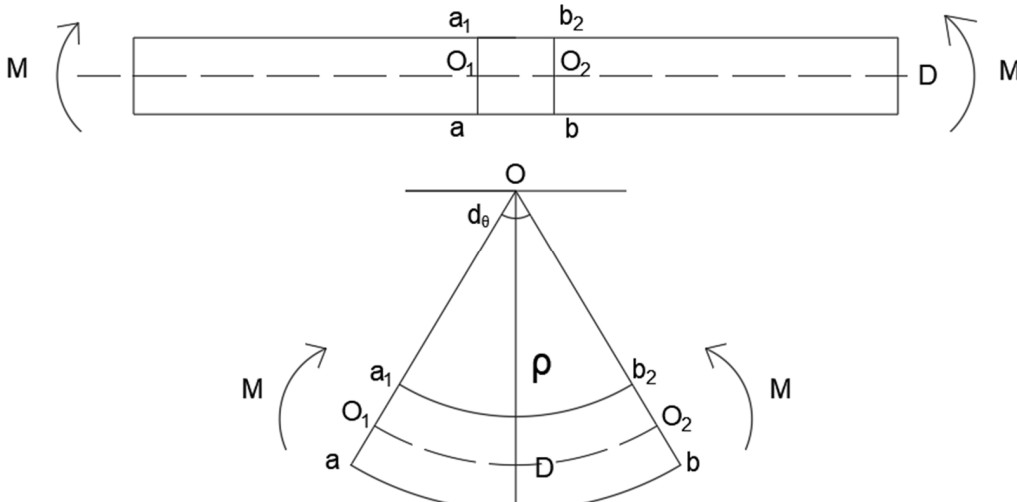

**Figure 3.** Schematic diagram of branch bending. M is the torque of the beam structure; O is the maximum deformation position of the beam structure; $O_1O_2$ is micro-element segment on neutral axis; ab is the elongated edge of the micro-element segment; $a_1b_2$ is the compressed edge of the micro-element segment; $\rho$ is the curvature radius of neutral layer after deformation; $D$ is the branch diameter.

The limit strain of plastic deformation of *Camellia oleifera* branch is shown in (5):

$$\varepsilon = \frac{\Delta s}{dx} = \frac{D}{2\rho} = \frac{DL^2}{24h} \tag{5}$$

*2.4. Research Methods*

2.4.1. Finite Element Analysis

The 3D model of the *Camellia oleifera* tree was established in SolidWorks and imported into ANSYS 19.2. The mesh generation is an important and critical step for the finite element analysis. Due to the irregularity of the geometric model of *Camellia oleifera*, solid185 was selected as the grid element for the simulation, and the intelligent dimension tool of ANSYS free-grid method was used for grid generation. The finite element model of the *Camellia oleifera* tree is regarded as a cantilever beam [32], in which the root of *Camellia oleifera* is fixed. In this study: (1) Modal analysis was carried out to determine the natural frequency of the camellia oil and describe the dynamic characteristics and behavior of the system; (2) The harmonic response analysis was carried out, and the external load was set as displacement to determine the steady response of the *Camellia oleifera* tree under a vibration load; (3) Through transient analysis, the actual vibration response of the *Camellia oleifera* tree under an external load was determined.

In this study, we used the block LANCZOS method for modal analysis. The X, Y and Z directions of the root are defined as the zero displacement constraints. In addition, the natural frequencies and modal shapes of *Camellia oleifera* were determined. The complete method was proved to be better for the harmonic response analysis of complex trees [33]. The complete method uses the complete system matrix to calculate the harmonic response (without matrix reduction), and uses a single process to calculate the displacement and stress. In the harmonic response analysis, the frequency range is determined by the modal analysis results. Considering the measurement range of the acceleration sensor, in the transient analysis, the simulation time is set to 1 s, and the amplitude and frequency loading are shown in Table 1.

**Table 1.** Amplitude and frequency combinations used in transient simulation and testing.

| NO. | Amplitude (mm) | Frequency (Hz) |
|---|---|---|
| 1 | 30 | 7 |
| 2 | 30 | 5 |
| 3 | 40 | 5 |

2.4.2. Vibration Test

The test of the *Camellia oleifera* tree adopted the canopy vibration-type *Camellia oleifera* fruit-picker developed by the research group. The vibration frequency and amplitude of the picker were adjustable. The vibration frequency adjustment range was 0~25 Hz, and the amplitude adjustment ranges were 30, 40, 50, 60 and 70 mm, respectively. The vibration position was consistent with the harmonic response analysis. We obtained the following for the test: dynamic signal acquisition and analysis system (KDDASP; Yangzhou Kedong Electronics Co., Ltd., Yangzhou, China); multi-functional static strain gauge (KD7016A; Yangzhou Kedong Electronics Co., Ltd., Jiangsu, China); accelerometer (DH311E; Donghua Detection Technology Co., Ltd., Taizhou, China); SA-LC02K hammer (Wuxi Shiao Technology Co., Ltd., Wuxi, China); disposable strain gauge. The dynamic test system was composed of a computer with analysis software(MATLAB R2020b; MathWorks, Natick, MA, USA). The sensors were fixed on the tree with adhesive tape, and their directions were random; the strain gauge was attached to the epidermis of *Camellia oleifera* tree with special glue, as shown in Figure 4.

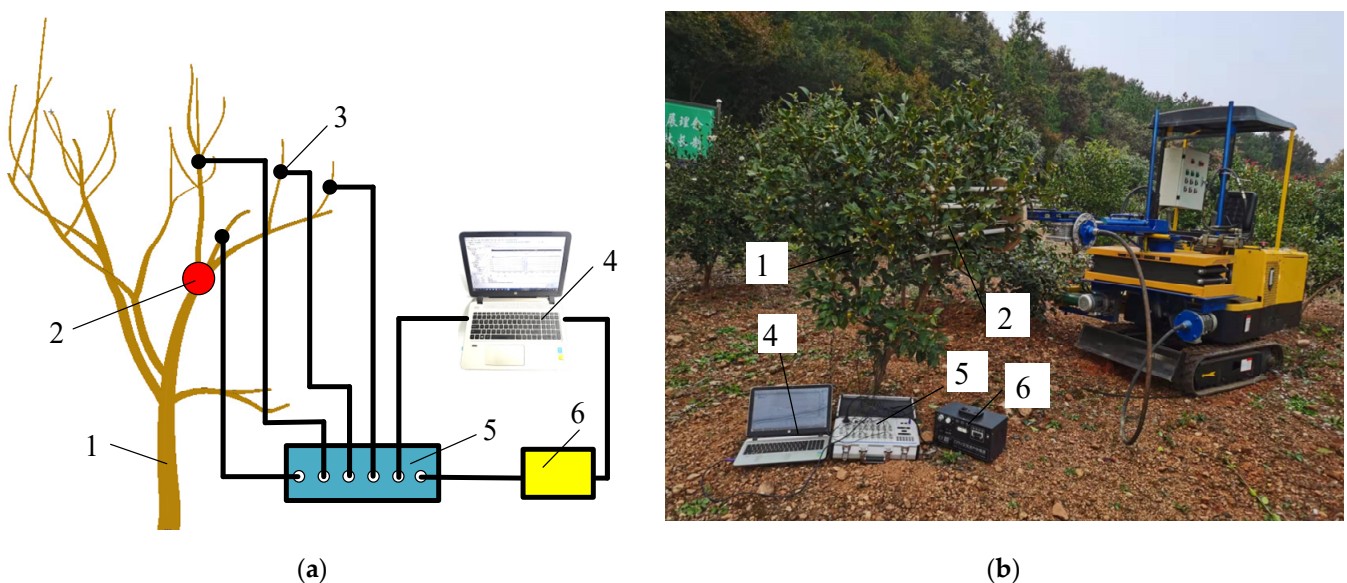

**(a)**                                                                 **(b)**

**Figure 4.** (**a**) Schematic diagram of data acquisition; (**b**) Test diagram of data acquisition. 1. *Camellia oleifera* tree; 2. Excitation position; 3. Acceleration sensor and strain gauge; 4. Computer; 5. Signal acquisition instrument; 6. Portable power source.

The main measurement methods were as follows: First, the acceleration sensor was fixed to the measured position of the tree, as shown in Figure 2, and the measurement position spacing was about 0.2 m. When the *Camellia oleifera* fruit-picker was used for vibration, the data were recorded by the data collector and the corresponding software. Secondly, the acceleration sensor was fixed at the next measurement position for the next measurement after the recording the data. Lastly, following the same procedure as above until all of the positions on the whole tree were measured. Similarly, the strain measurement method was the same. The strain was measured by the acceleration measurement method.

## 3. Results and Discussions

### 3.1. Physical Characteristics of Camellia Oleifera

The physical parameters of the *Camellia oleifera* tree were measured according to the above method, as shown in Table 2.

**Table 2.** Physical characteristics of *Camellia oleifera* tree.

| Materials | Elasticity Modulus (Mpa) | Density (g/cm$^3$) | Damping Ratio | Poisson's Ratio |
|---|---|---|---|---|
| *Camellia oleifera* tree | 326.66 | 0.95 | 0.06 | 0.3 [34] |

The 15 groups of test data were fitted by cubic polynomial. The relationship between the *Camellia oleifera* branch diameter and ultimate strain value is shown in Figure 5.

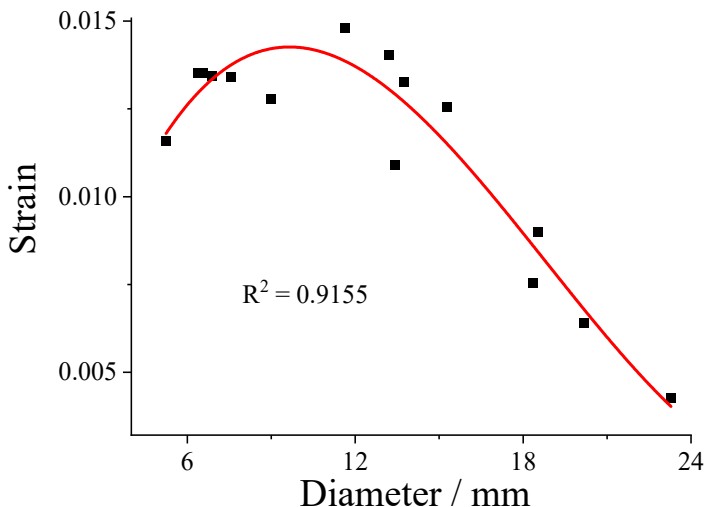

**Figure 5.** Relationship between *Camellia oleifera* branch diameter and ultimate strain value. $R^2$ is the polynomial fitting correlation coefficient.

The relationship between the strain of *Camellia oleifera* branch during plastic deformation and the branch diameter is shown in (6):

$$Y = 5.23 + 0.003X - 2.24X^2 + 3.98X^3 \tag{6}$$

where $Y$ is plastic deformation; $X$ is the branch diameter.

As can be seen from Figure 5, the $R^2$ of the fitting equation is 0.9155, indicating that this equation can reflect 91.55% of the response value changes, indicating that the polynomial regression equation model obtained has a high degree of fitting and a good fitting effect. The maximum plastic strain of the *Camellia oleifera* branches was 0.0148, and the diameter of *Camellia oleifera* tree was 11.62 mm. On the other hand, there was a nonlinear correlation between the plastic strain value and the branch diameter. When the diameter of the *Camellia oleifera* branch was less than 9 mm, the plastic strain value of the branch increased with the increase in the branch diameter. When the diameter of *Camellia oleifera* branch was greater than 9 mm, the plastic strain value of the branch decreased first and then increased with the increase in the diameter of the branch.

### 3.2. Free Mode Analysis of Camellia Oleifera

The modal module in the analysis workbench 19.2 was used to analyze the free mode of *Camellia oleifera* model at order 1~50, and the relationship between the frequency and shape variable under free mode of the *Camellia oleifera* tree model is shown in Table 3.

**Table 3.** Relationship between frequency and shape variables under free mode of *Camellia oleifera* model.

| No. | Frequency (Hz) | Maximum Response Point | No. | Frequency (Hz) | Maximum Response Point | No. | Frequency (Hz) | Maximum Response Point |
|---|---|---|---|---|---|---|---|---|
| 1 | 1.67 | c1 | 18 | 5.41 | b2 | 35 | 11.68 | a2 |
| 2 | 1.94 | a6 | 19 | 5.85 | a1 | 36 | 12.01 | a1 |
| 3 | 2.78 | a1 | 20 | 6.29 | a5 | 37 | 12.37 | b4 |
| 4 | 2.95 | b5 | 21 | 6.73 | a4 | 38 | 12.63 | b2 |
| 5 | 3.09 | c3 | 22 | 7.21 | a1 | 39 | 12.77 | a1 |
| 6 | 3.33 | b2 | 23 | 7.48 | a3 | 40 | 12.78 | a2 |
| 7 | 3.48 | c4 | 24 | 7.63 | b2 | 41 | 12.84 | a2 |
| 8 | 3.55 | b1 | 25 | 7.94 | c4 | 42 | 13.15 | b3 |
| 9 | 3.68 | c4 | 26 | 8.56 | c3 | 43 | 13.63 | c3 |
| 10 | 4.08 | c3 | 27 | 8.91 | a1 | 44 | 13.86 | c4 |
| 11 | 4.48 | a1 | 28 | 9.08 | a6 | 45 | 14.02 | a6 |
| 12 | 4.62 | a2 | 29 | 9.14 | b3 | 46 | 15.35 | b2 |
| 13 | 4.64 | b3 | 30 | 9.65 | c2 | 47 | 15.74 | a1 |
| 14 | 4.79 | c3 | 31 | 10.27 | a1 | 48 | 16.13 | a6 |
| 15 | 4.96 | a2 | 32 | 10.43 | c4 | 49 | 16.68 | c3 |
| 16 | 5.11 | a1 | 33 | 11.02 | c4 | 50 | 17.18 | a6 |
| 17 | 5.34 | c2 | 34 | 11.10 | c3 | | | |

Note: $a_i$, $b_i$, $c_i$, $d_i$ represent different second-level branches of *Camellia oleifera*.

Because the position of the maximum deformation in free mode analysis is not fixed, and the maximum deformation obtained is not at the same position, it is difficult to draw a conclusion and data processing is required. The frequency is rounded off, and the shape variable of the corresponding frequency band is taken as the average value. The processed data can reflect the average shape variable in a certain frequency band, which is representative. The broken line diagram of different frequency bands and shape variables is shown in Figure 6.

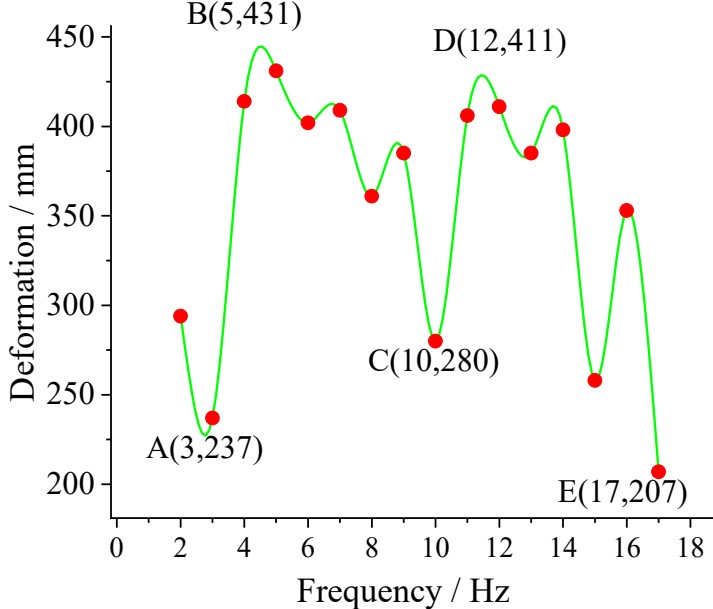

**Figure 6.** Line chart of frequency and deformation response of *Camellia oleifera*. A, B, C, D and E represent the excitation frequencies with typical displacement characteristics.

It can be seen from Figure 6 that under the free vibration state of the *Camellia oleifera* tree model, the shape variable is divided into two sections as a whole. The boundary frequency

is 10 Hz, the first section is 2~10 Hz, and the second section is 10~17 Hz. There are maximum values in each section, which are 431 and 411 mm, respectively. The maximum values appear in the frequency bands of 5 and 12 Hz, respectively. The minimum values of the average shape variables are 237, 280 and 207 mm, respectively, which appear in the frequency bands of 3, 10 and 17 Hz, respectively.

Through the comparative analysis of the free-mode nephogram with typical characteristics at A, B, C, D and E, as shown in Figure 7, it can be seen that the model response at the B, C and D frequency bands has a better overall response effect, a higher canopy response and a lower trunk response from the perspective of the overall response effect; the overall response effect of the model response in the A and E frequency bands is not very ideal, and only a few branches of the canopy respond. From the point of view of the shape variables: the model response in the A, B and C frequency bands has small shape variables, which will not cause great damage to the tree. The response of the model in the D and E frequency bands has large shape variables, and the overall structure of the tree is easily damaged when it vibrates at this frequency band. The *Camellia oleifera* fruits are concentrated at about 260 mm away from the crown surface [35], and only the crown vibration is required for actual harvest. Therefore, the optimal frequency range in the vibration harvesting process of *Camellia oleifera* can be controlled between 4~10 Hz.

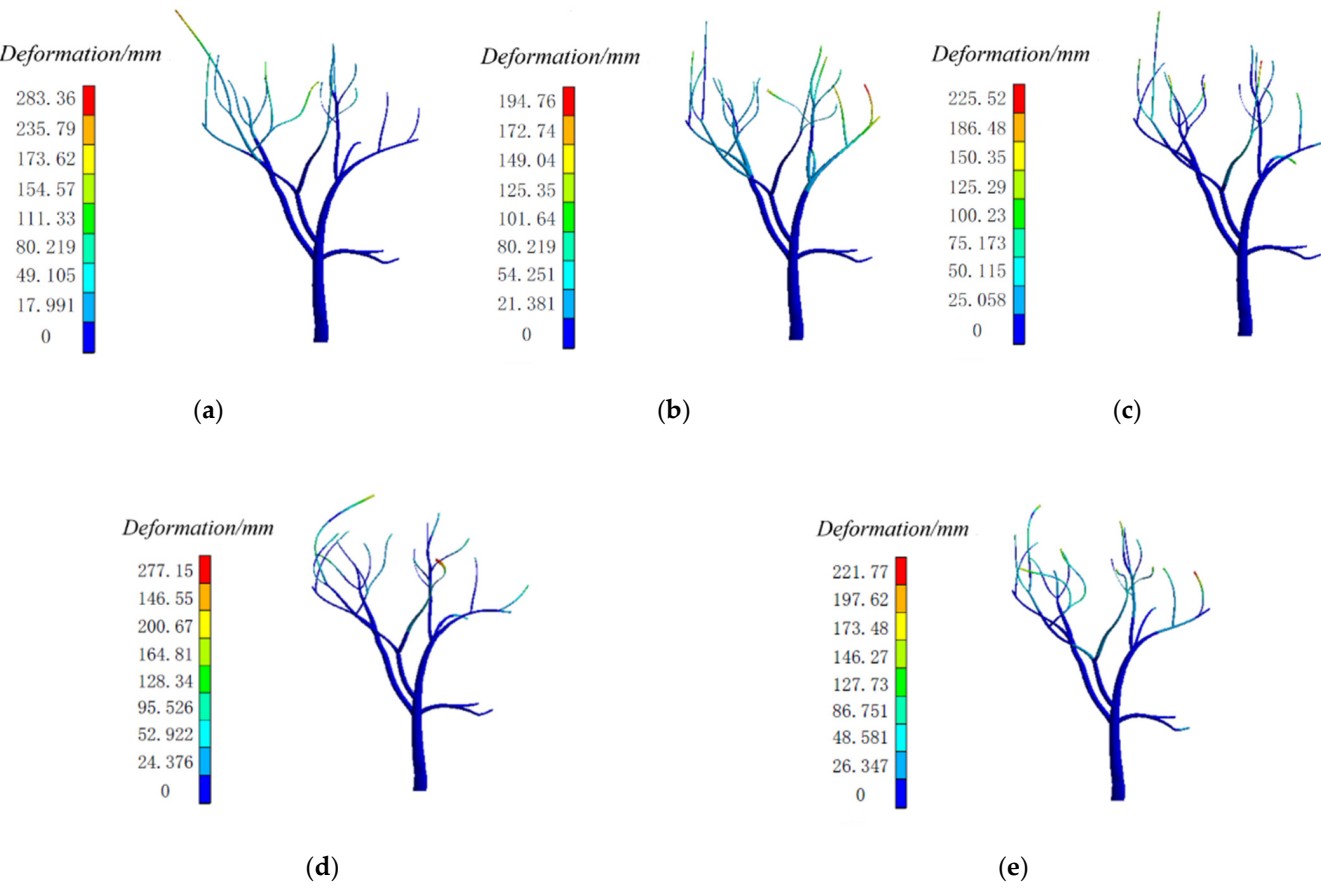

**Figure 7.** Model of free-response cloud image comparison. (**a**) Response cloud map representing point A of typical displacement feature; (**b**) Response cloud map representing point B of typical displacement feature; (**c**) Response cloud map representing point C of typical displacement feature; (**d**) Response cloud map representing point D of typical displacement feature; (**e**) Response cloud map representing point E of typical displacement feature.

### 3.3. Harmonic Response Analysis of Camellia Oleifera

3.3.1. Analysis of Shedding Conditions of *Camellia oleifera* Fruit

The resonance between the tree and the applied load is the key factor of the vibration collection, because it determines the separation of the *Camellia oleifera* fruits and branches. Figure 8 shows the stress diagram of the *Camellia oleifera* fruits. According to Newton's second law, Equation (7) can be obtained:

$$F + mg - F_L = ma \tag{7}$$

where $F$ is the vibration load; $F_L$ is the binding force of the branch to the fruit; $m$ is the weight of the fruit; $a$ is the acceleration; and $g$ is the acceleration caused by the earth's gravity.

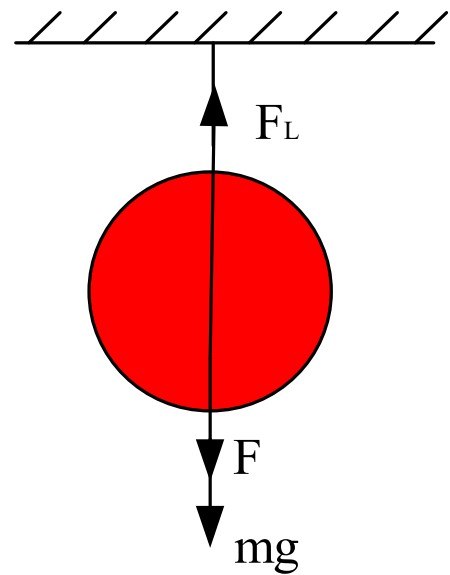

**Figure 8.** Stress analysis of *Camellia oleifera* fruits. $F$ is the vibration load; $F_L$ is the binding force of the branch to the fruit; $m$ is the weight of the fruit; $a$ is the acceleration; and $g$ is the acceleration caused by the earth's gravity.

According to the previous studies, the average weight of the *Camellia oleifera* fruit is 28 g and the maximum binding force is 18 N. According to Equation (4), the minimum acceleration of the *Camellia oleifera* fruit falling off is about 427 m/s².

3.3.2. Harmonic Response Analysis of Canopy under Stress

The harmonic response analysis determines the response of *Camellia oleifera* tree under different frequencies, and finds the best corresponding frequency. According to the free mode analysis of the *Camellia oleifera* tree, the optimal excitation frequency is 4~10 Hz during the mechanized vibration-picking of *Camellia oleifera*. Therefore, the frequency range of 4~10 Hz is set during the harmonic response analysis.

The stress at point P of the *Camellia oleifera* canopy and the marked positions of the eight selected points are shown in Figure 9a. When the displacement load at point P is 10 mm, the acceleration response at the eight selected points is shown in Figure 9b. There are multiple peaks, including two obvious peaks. The peaks at the detection points on the different branches are inconsistent. The acceleration responses of test points two and six reach 92 and 100 m/s², respectively, at 9 Hz, which are relatively large. In addition, the acceleration responses of test points six and seven reach 62 and 50 m/s² at 5 Hz, which are also large.

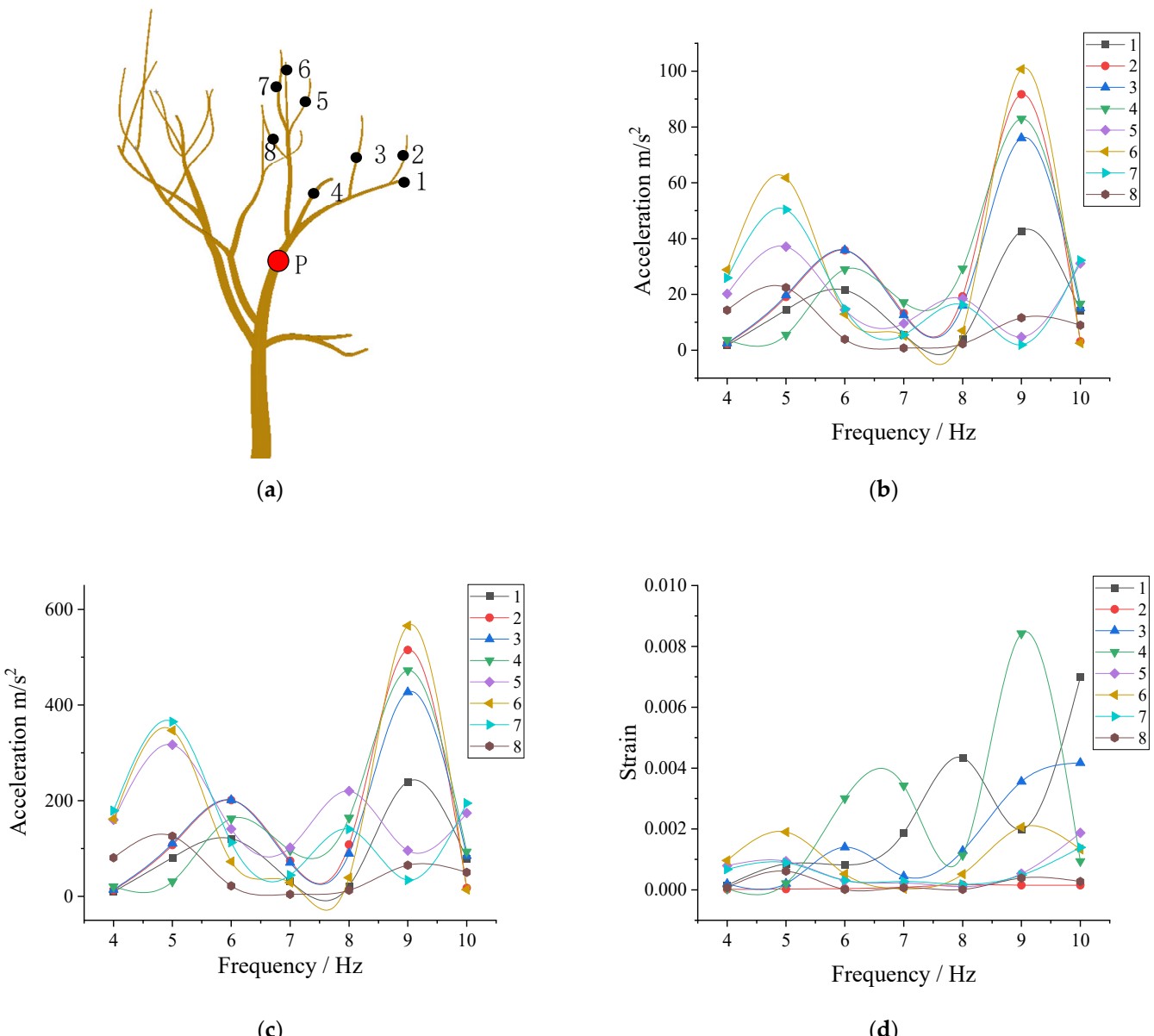

**Figure 9.** Harmonic response analysis of *Camellia oleifera* canopy under stress. (**a**) 8 representative points when the vibration position is point P; (**b**) Acceleration response of 8 representative points when 10 mm displacement load is applied to the canopy; (**c**) Acceleration response of 8 representative points when 56 mm displacement load is applied to the canopy; (**d**) Strain response of 8 representative points under 56 mm displacement load on canopy.

The detection points two, six and seven are on different branches, detection point two is on the right branch, and detection points six and seven are on the left branch; this shows that the optimal frequency of vibration is related to the branch position. The maximum acceleration response of detection points three and four appears at 9 Hz, which are 76 and 83 m/s², respectively. Therefore, appropriate force should be applied to ensure that all of the fruits at test points three and four can be separated. The harmonic response analysis can be regarded as a linear analysis [36] and can be expressed as Equation (8):

$$\frac{A_1}{a_1} = \frac{A_2}{a_2} \tag{8}$$

where $A_1$ is the applied 10 mm vibration force; $a_1$ is the acceleration response value corresponding to the detection point under the 10 mm vibration force; $A_2$ is the best vibration force; and $a_2$ is the acceleration required to separate the fruit from the branch.

Therefore, detection point three obtained a large acceleration response at 9 Hz. As shown in Figure 9b, according to the linear characteristics of the harmonic response analysis, when the acceleration response at detection point three reached 427 m/s$^2$, the displacement load required for point P was 56 mm. A displacement load of 56 mm was applied at point P, and the harmonic response analysis acceleration response of each detection point is shown in Figure 9c. Comparing Figure 9b,c, it is obvious that the acceleration response of each detection point increases with the increase in the displacement load at point P, and the change trend does not alter. When a displacement load of 56 mm is applied to point P and the excitation frequency is 9 Hz, the acceleration response of detection points two–four and six is greater than 427 m/s$^2$. However, the acceleration response of the other detection points is less than 427 m/s$^2$.

When a displacement load of 56 mm is applied to point P, the strain of each detection point in the harmonic response analysis is shown in Figure 9d. The strain value of detection point four is 0.0084 when the excitation frequency is 9 Hz, which is relatively large. The strain value of detection point one is 0.0070 when the excitation frequency is 10 Hz, which is also very large. However, when the excitation frequency is 9 Hz, the strain values of the other test points, except test point four, are relatively small, and the maximum value at test point three is 0.0036. After measurement, the diameter of each detection point on the *Camellia oleifera* branch is between 5~12 mm. According to the relationship between the strain of the *Camellia oleifera* branch during plastic deformation and the branch diameter, the limit strain value is 0.012 when the diameter of the *Camellia oleifera* branch is between 5~12 mm. Therefore, when the excitation frequency is 9 Hz for vibration picking, the *Camellia oleifera* branches will not undergo plastic deformation.

### 3.4. Comparison between Transient Analysis and Test Measurement Results

According to the location of the detection point shown in Figure 9a, the acceleration sensor and strain gauge are arranged. The real test is carried out on the *Camellia oleifera* tree, and the responses of the *Camellia oleifera* tree under the different collective conditions are collected. At the same time, transient analysis of Ansys was used to simulate the *Camellia oleifera* tree. Finally, the test and simulation results are compared.

### 3.4.1. Acceleration Response of Each Detection Point

The actual vibration is counted by the composite value of acceleration, that is, the vector sum of three measuring axes on each acceleration sensor is used for vibration analysis. The formula for calculating the resultant acceleration value is shown in (9):

$$a = \sqrt{a_x^2 + a_y^2 + a_z^2} \tag{9}$$

where $a$ is the resultant acceleration; $a_x$, $a_y$ and $a_z$ are the acceleration values in the X, Y and Z directions, respectively.

It can be seen from Figure 10 that the acceleration in the test is close to the simulation value. It can be seen from the figure that most points have similar trends, but a few points have different trends. Under the different excitation parameters, the maximum acceleration occurs at the detection point two. Compared with Figure 10a,b, it is found that the acceleration response of the different detection points is not linearly correlated with the frequency, that is, the acceleration response of the different branches is non-linearly correlated with the frequency; comparing Figure 10b with Figure 10c, it is found that the overall change trend is basically unchanged, but the acceleration value is different. Therefore, the acceleration response of each detection point is correlated with the amplitude, that is, the acceleration response of each branch increases with the increase in the amplitude.

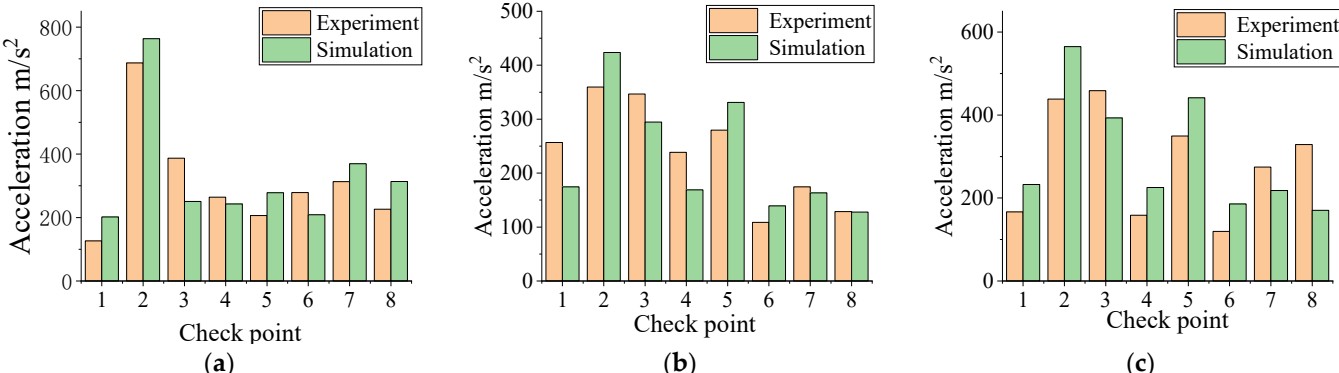

**Figure 10.** Comparison of acceleration response between transient analysis and test results. (**a**) 30 mm, 7 Hz; (**b**) 30 mm, 5 Hz; (**c**) 40 mm, 5 Hz.

### 3.4.2. Strain at Each Test Point

It can be seen from Figure 11 that the strain in the test is close to the simulation value. It can be seen from the figure that most of the points have similar trends, but a few points have different trends. Under different excitation parameters, the maximum strain appears at the detection point two. Compared with Figure 11a,b, it is found that the strain at the different detection points is not linearly correlated with the frequency, that is, the acceleration response of the different branches is non-linearly correlated with the frequency; comparing Figure 11b with Figure 11c, it is found that the overall change trend of the analog value basically does not change, only the difference of the acceleration value. Therefore, the acceleration response of each detection point is correlated with the amplitude, that is, the acceleration response of each branch increases with the increase of amplitude.

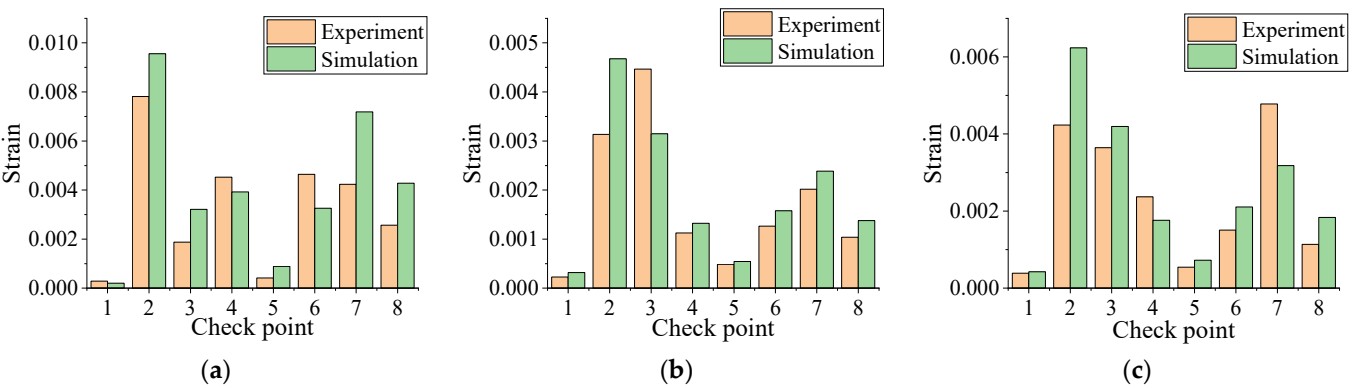

**Figure 11.** Comparison of strain response between transient analysis and test results. (**a**) 30 mm, 7 Hz; (**b**) 30 mm, 5 Hz; (**c**) 40 mm, 5 Hz.

### 3.4.3. Correlation Coefficient Analysis

Analyzing the correlation between the test value and fitting value can further elucidate the accuracy of the simulation value. The correlation coefficient between the simulation value and the test measurement value is shown in Figure 12. When the frequency is 5 Hz and the amplitude is 30 mm, the maximum correlation coefficient of the acceleration response is 0.89, it shows that the simulation analysis results can represent 89% of the real test results; when the frequency is 5 Hz and the amplitude is 40 mm, the minimum correlation coefficient of the acceleration response is 0.74, it shows that the simulation analysis results can represent 74% of the real test results. When the frequency is 7 Hz and the amplitude is 30 mm, the maximum correlation coefficient of strain is 0.89, this shows that the simulation analysis results can represent 89% of the real test results; when the frequency is 5 Hz and the amplitude is 40 mm, the minimum correlation coefficient

of strain is 0.83, this shows that the simulation analysis results can represent 83% of the real test results.

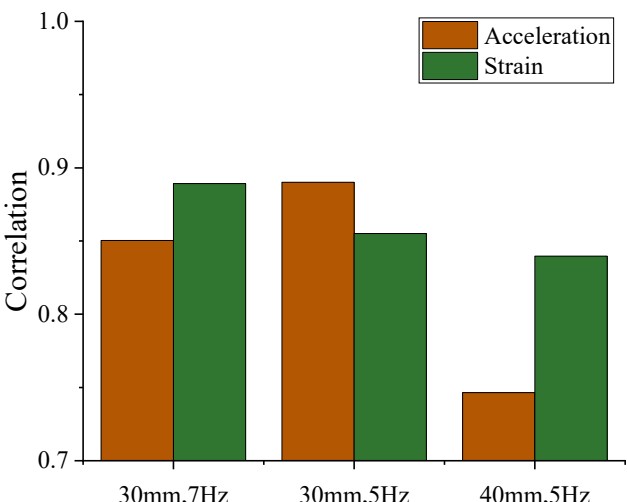

**Figure 12.** Data correlation coefficients of acceleration and strain between test and simulation under different parameter combinations.

Therefore, the simulation analysis results can basically represent the actual test results through certain coefficient calculations.

### 3.4.4. Curve Fitting Analysis

We studied the corresponding relationship between the simulation and the test, and used MATLAB to fit the linear and quadratic curves of acceleration response and strain, as shown in Equations (10) and (11):

$$a_c = p_1 + p_2 a_m \tag{10}$$

$$a_c = p_1 + p_2 a_m + p_3 a_m^2 \tag{11}$$

where $a_c$ is the test acceleration; $a_m$ is the simulation acceleration; $P_2$ and $P_3$ are the primary term coefficient and the secondary term coefficient, respectively, $P_1$ is the constant coefficient.

According to the data given in Table 4, the $R^2$ value after linear fitting adjustment of acceleration is greater than that of quadratic fitting curve. In other words, the interpretation rate of linear fitting simulation to the test results is 69%, which is higher than that of quadratic fitting by 67%. Therefore, the acceleration response of simulation and test results can be approximately linearly correlated. The adjusted $R^2$ values of the linear fitting and quadratic fitting curves of the strain are the same, both of which are 77%. Therefore, the strain of the simulation and the test results can be approximately linearly correlated, or can be considered as a quadratic function correlation.

**Table 4.** Fitting coefficients and adjusted $R^2$ under different parameters.

|  | Acceleration (m/s$^2$) | | Strain | |
| --- | --- | --- | --- | --- |
|  | Linear Fitting | Quadratic Fitting | Linear Fitting | Quadratic Fitting |
| $P_1$ | 72.28 | 81.12 | 0.00033 | 0.000024 |
| $P_2$ | 0.71 | 0.66 | 0.74 | 0.98 |
| $P_3$ | / | 0.000068 | / | −28.44 |
| $R^2$ | 0.69 | 0.67 | 0.77 | 0.77 |

Note: $P_1$ is the constant coefficient; $P_2$ and $P_3$ are the primary term coefficient and the secondary term coefficient, respectively; $R^2$ is the polynomial fitting correlation coefficient.

### 3.5. Test Verification

3.5.1. Verifying Harmonic Response Analysis Results

A simple harmonic load with a frequency of 9 Hz and an amplitude of 56 mm is applied to point P (Figure 9a). The acceleration response and maximum strain values of each detection point are shown in Figure 13, through transient analysis. The maximum acceleration of test point six in the eight selected points is 710 m/s$^2$, and the maximum acceleration of the other test points is greater than 427 m/s$^2$, as shown in Figure 13a. The accelerations of these points are substituted into the linear fitting results. The accelerations of test points two, five and six are greater than 427 m/s$^2$, and the accelerations of the other test points are at least 382 m/s$^2$. They all reach the acceleration required for fruit dropping. Therefore, when the vibration parameters are 9 Hz and 56 mm, the *Camellia oleifera* fruits can fall off at the detection points two, five and six6. This parameter is suitable for the picking of the *Camellia oleifera* fruits.

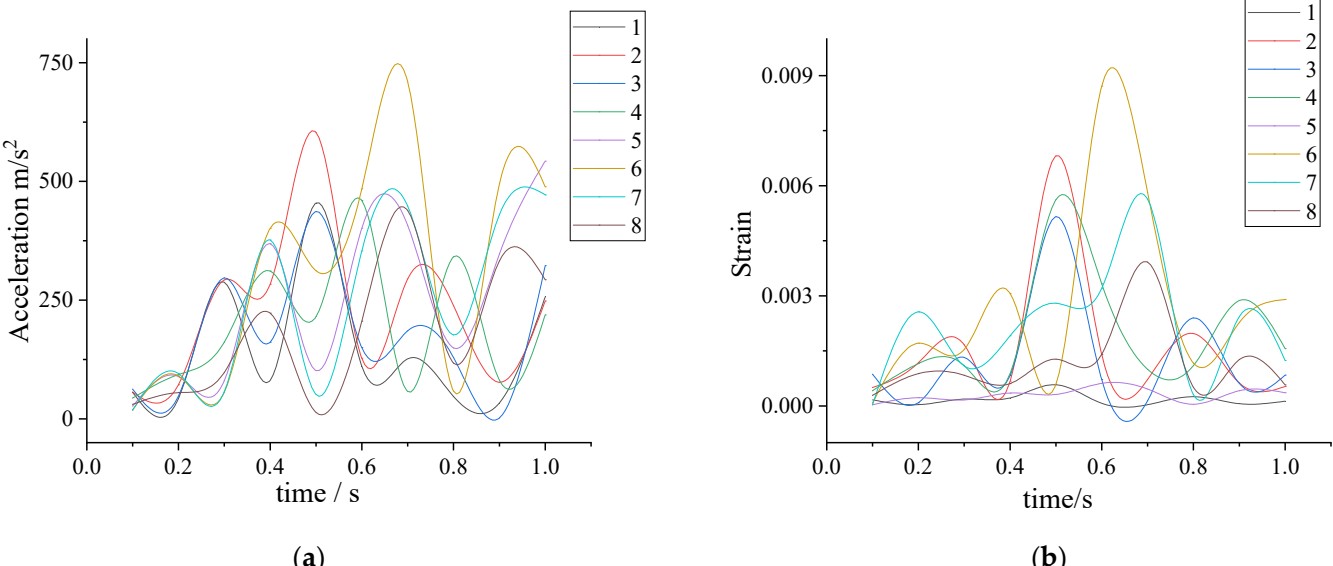

(**a**)  (**b**)

**Figure 13.** Transient analysis results of each detection point. (**a**) is the transient analysis of acceleration response; (**b**) is the transient analysis of strain response.

The maximum strain of test point six in the eight selected points is 0.0087, and the maximum strain of the other test points is less than the maximum strain of test point six. Therefore, the maximum strain of all of the test points is less than 0.012, as shown in Figure 13b. By substituting the strain values of these points into the linear fitting results, the maximum strain value of each detection point in the corresponding test is 0.0068. Therefore, when the vibration parameters are 9 Hz and 56 mm, the branches at each detection point will not undergo plastic deformation.

3.5.2. Vibration Picking of *Camellia oleifera* Tree Canopy

Due to the limited control accuracy of the *Camellia oleifera* fruit-picker, a simple harmonic load with a frequency of 9 Hz and an amplitude of 60 mm was applied to the *Camellia oleifera* tree canopy. According to the basis of the previous research [37] the excitation time was selected as 5, 10 and 15 s for field test. Before the vibration test, the number of *Camellia oleifera* fruits and flower buds were counted, and after the vibration, the number of fallen fruits and flower buds were counted. The picking of the *Camellia oleifera* fruits is shown in Table 5.

**Table 5.** Mechanized picking of *camellia oleifera*.

| Index | 5 s | | 10 s | | 15 s | |
|---|---|---|---|---|---|---|
| | **Mean** | **Standard Deviation** | **Mean** | **Standard Deviation** | **Mean** | **Standard Deviation** |
| Fruit shedding rate (%) | 68 | 11 | 90 | 5 | 91 | 5 |
| Bud abscission rate (%) | 11 | 3 | 13 | 4 | 18 | 6 |

Note: Data in the table are from 30 samples.

As shown in Table 3, when picking *Camellia oleifera* fruits with a simple harmonic force with a frequency of 9 Hz and an amplitude of 60 mm, the maximum average abscission rate of *Camellia oleifera* fruits is 91%, and the excitation time is 15 s. Therefore, the frequency of 9 Hz and amplitude of 60 mm can meet the requirements for the mechanized vibration-picking of the *Camellia oleifera* fruits.

## 4. Discussion

The factors affecting the mechanized vibration-picking effect of *Camellia oleifera* fruits are not only determined by the frequency and amplitude [17,22], but also by the vibration time [20]. With the increase in the vibration time, the fruit abscission rate of *Camellia oleifera* also increased. When the vibration time was 15 s, the maximum fruit abscission rate of *Camellia oleifera* was 91%. However, when the excitation time was increased from 5 s to 10 s, the fruit abscission rate of the *Camellia oleifera* increased by 24%, and when the excitation time was increased from 10 s to 15 s, the fruit abscission rate of *Camellia oleifera* increased by 1%. Therefore, from the perspective of the drop rate of the *Camellia oleifera* fruits, the vibration time of 10 s or 15 s can achieve a better mechanized picking-effect of the *Camellia oleifera* fruits. At the same time, with the increase in excitation time, the flower bud abscission rate of *Camellia oleifera* also increased. When the vibration time was 5 s, the minimum flower bud abscission rate was 11%. However, when the excitation time was increased from 5 s to 10 s, the flower bud abscission rate of *Camellia oleifera* increased by 15%, and when the excitation time was increased from 10 s to 15 s, the flower bud abscission rate of *Camellia oleifera* increased by 28%. Therefore, from the perspective of flower bud-falling rate of *Camellia oleifera*, a vibration time of 5 s or 10 s can obtain a better mechanized picking-effect of *Camellia oleifera*. Combined with the falling off of the *Camellia oleifera* fruits and flower buds during the vibration picking process, the vibration time of 10 s is better for the mechanized picking of *Camellia oleifera*.

Under the different objective functions, the vibration time, as well as the vibration frequency and amplitude, also has the optimal solution [37].Of course, if the vibration time is prolonged, we have reason to believe that the fruit abscission rate of *Camellia oleifera* will not increase significantly, because when the excitation time is increased from 10 s to 15 s, the fruit abscission rate of *Camellia oleifera* will only increase by 1%, but the flower bud abscission rate of *Camellia oleifera* will increase by 28%.

During the vibration picking of *Camellia oleifera* fruits, the fruits did not completely fall off [23]. First, the physical parameters related to *Camellia oleifera* used in the simulation were all from the average values. The unswept fruits are mostly the green fruits or immature fruits. The binding force of the immature *Camellia oleifera* fruits is greater than that of the mature fruits. The *Camellia oleifera* fruits cannot obtain enough inertia force during the vibration process, resulting in the *Camellia oleifera* fruits not falling off. This paper did not fully consider the impact of the different maturity of *Camellia oleifera* fruits on the mechanized picking–effect. From the field test results, it can be seen that the different maturity of the *Camellia oleifera* fruits will have a certain impact on the picking effect. The next step of this study will focus on the role of the *Camellia oleifera* fruits of a different maturity in the process of the mechanized picking the impact on *Camellia oleifera* of the mechanized picking. Therefore, this study presents a subject to be further studied.

**5. Conclusions**

(1) The 3D model of the *Camellia oleifera* tree was established, and the free modal analysis of *Camellia oleifera* tree was carried out by ANSYS. It was determined that the optimal vibration frequency range for the *Camellia oleifera* tree was 4~10 Hz. The modal analysis determined that the optimal excitation frequency and amplitude of the *Camellia oleifera* tree were 9 Hz and 56 mm, respectively. The results show that the acceleration responses and strains in the test are close to the simulation value, but there are some errors. The correlation coefficient between the test value and the simulation value is 0.89;

(2) When the frequency is 9 Hz and the amplitude is 56 mm, the acceleration of the detection points on the *Camellia oleifera* branches is greater than the acceleration required for the fruits to fall off, and the strain values of each detection point are less than the plastic deformation condition, that is, the *Camellia oleifera* tree will not undergo plastic deformation under this vibration parameter;

(3) The best vibration parameters were further verified through field experiments. The test results showed that the vibration frequency of 9 Hz, the amplitude of 60 mm and the vibration time of 10 s were more suitable for the mechanized picking of *Camellia oleifera*. At this time, the fruit abscission rate of the *Camellia oleifera* was 90%, and the flower bud damage-rate was 13%. In general, this study can provide guidance for the production of harvesting machinery and improve the harvesting efficiency.

**Author Contributions:** Conceptualization, D.W. and W.W.; data curation, E.Z., S.J., C.W. and R.W.; formal analysis, D.F. and W.W.; investigation, E.Z., S.J. and C.W.; Methodology, D.W. and E.Z.; project administration, D.W., S.J. and W.W.; software, C.W.; supervision, S.J.; validation, D.F.; visualization, D.F. and R.W.; writing–original draft, E.Z. and C.W.; writing–review and editing, D.W., D.F. and R.W. All authors have read and agreed to the published version of the manuscript.

**Funding:** This research was funded by the Natural Science Foundation of Anhui Province (NO.2208085ME132) and the National Key Research and Development Program of China (NO.2016YFD0702105).

**Institutional Review Board Statement:** Not applicable.

**Informed Consent Statement:** Not applicable.

**Data Availability Statement:** The data used in this study are of self-test and self-collection. Because this research direction still needs further development and improvement, the data cannot be shared at present.

**Acknowledgments:** The authors would like to thank the editors and all the reviewers who participated in the review.

**Conflicts of Interest:** The authors declare no conflict of interest.

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
