# Peer review of "Determination of Vibration Picking Parameters of Camellia oleifera Fruit Based on Acceleration and Strain Response of Branches"

_agriculture, doi:10.3390/agriculture12081222_

Round 1
Reviewer 1 Report
Substantive assessment:
Using the vibration method, the authors determined the optimal parameters of seed harvesting from fairly tall Camellia oleifera trees. It is a plant little known outside of Asia, hence the interest in this method seems to be low. On the other hand, there are many similar seeds in the shell that can be harvested by the proposed technique without damaging delicate branches and inflorescences.
The vibration method itself during harvesting is not new. Gently shaking off trees and shrubs has been around for decades. There are also machines for such technology, which of course can and should be modified all the time. However, the greatest added value of this manuscript is the presented methodology, which uses modern computer simulation techniques (FEM, Solidworks). Especially the 3D model of the plant (i.e. the artifact) deserves recognition. Additionally, it validates this method on the example of Camella oleifera vibratory harvesting. Thus, it makes the whole valuable for science and practice.
Nothing is mentioned about the reliability of the machines, which are as important as their performance and failure rate. Examples of methods for quantifying the reliability of agricultural machinery are included, for example, in the Agronomy MDPI publishing house:
Durczak, K.; Selech, J.; Ekielski, A.; Å»elaziÅ„ski, T.; WaleÅ„ski, M.; Witaszek, K. Using the Kaplan–Meier Estimator to Assess the Reliability of Agricultural Machinery. Agronomy 2022, 12, 1364. https://doi.org/10.3390/agronomy12061364
Such field or accelerated reliability tests on measuring stands should be the next stage of activities of this team of authors.
Editorial assessment:
In scientific works (also in their titles), the names of plants (and animals) are also given in the original version, i.e. in Latin, and then in italics.
According to the rules of the SI system, there is to be a gap between the value of the measurement result and the unit. They are missing, for example, in the Summary and in the main text, e.g. in lines 115, 192, 193, 210, 245, ...
I find giving index values ​​up to 0.01% too accurate. The indicator itself is already burdened with many errors. So instead of a 12.74% damage rate, 13% is enough and a fruit cut-off rate instead of 89.92% is enough for 90%. It also allows the reader to quickly analyze them (is a lot or a little?). The same remark applies to values ​​for N (e.g., line 281) or m/s2 (line 282).
Check in the editorial requirements how the headings of the subsections are written (upper or lower case?).
Figure 1 caption should be specific to the content of a) and b). The same is true for Figure 2.
Correct Equation 1 (missing gaps).
Figure 2a poor quality.
In line 126 the parameter designations must comply with the formula.
The most important achievements (i.e. conclusions) could be listed.
The original way of citing a bibliography (rather Latin numerals).
Author Response
Author Response 1
Dear peer reviewers and editors
Hello! Thank you very much for your professional and wise comments on this article. According to the experts' questions and opinions, the author gives the detailed explanations in the form of one question and one answer, and marks the manuscript with red font in the corresponding position, as follows:
QUESTION1: In scientific works (also in their titles), the names of plants (and animals) are also given in the original version, i.e. in Latin, and then in italics.
RE: Thank you very much for your valuable comments. The author changed the writing format of the name "camellia oleifera" and marked them in red.
QUESTION2: According to the rules of the SI system, there is to be a gap between the value of the measurement result and the unit. They are missing, for example, in the Summary and in the main text, e.g. in lines 115, 192, 193, 210, 245, ...
RE: Thank you very much for your valuable comments. The author added the one space between the numerical value and the measurement unit and marked them in red.
QUESTION3: I find giving index values ​​up to 0.01% too accurate. The indicator itself is already burdened with many errors. So instead of a 12.74% damage rate, 13% is enough and a fruit cut-off rate instead of 89.92% is enough for 90%. It also allows the reader to quickly analyze them (is a lot or a little?). The same remark applies to values ​​for N (e.g., line 281) or m/s2 (line 282).
RE: Thank you very much for your valuable comments. The authors have reduced the accuracy of the index values so that the reader can quickly analyze them (are they many or few?) and marked them in red.
QUESTION4: Check in the editorial requirements how the headings of the subsections are written (upper or lower case?).
RE: Thank you very much for your valuable comments. The author has changed it to the correct format upon review and editorial request and marked them in red.
QUESTION5: Figure 1 caption should be specific to the content of a) and b). The same is true for Figure 2.
RE: Thank you very much for your valuable comments. The author has changed the image title and marked them in red.
QUESTION6: Correct Equation 1 (missing gaps).
RE: Thank you very much for your valuable comments. The author has modified Equation 1 and marked them in red.
QUESTION7: Figure 2a poor quality.
RE: Thank you very much for your valuable comments. The author has replaced the pictures with higher quality one and marked them in red.
QUESTION8: In line 126 the parameter designations must comply with the formula.
RE: Thank you very much for your valuable comments. The author added parameter explanation according to the formula content and marked them in red.
QUESTION9: The most important achievements (i.e. conclusions) could be listed.
RE: Thank you very much for your valuable comments. The author sorted out and summarized the conclusion, and listed the main contributions and highlights separately and marked them in red.
QUESTION10: The original way of citing a bibliography (rather Latin numerals).
RE: Thank you very much for your valuable comments. The author consulted the guidelines for submission of the journal and downloaded several articles, and observed that the format of references cited was Arabic numerals. Such as (Wang, W.; Song, J.; Zhou, G.; Quan, L.; Zhang, C.; Chen, L. Development and Numerical Simulation of a Precision Strip-Hole Layered Fertilization Subsoiler While Sowing Maize. Agriculture 2022, 12, 938. https://doi.org/10.3390/agriculture12070938).
Thank you very much for your valuable opinions. Please review the revised article!

Reviewer 2 Report
Review one more time the spelling of letters in all text: title „Determination Of Vibration Picking“etc.
In general, references are numbered in Arabic numerals in the texts. „...reference numbers should be placed ...before the punctuation; for example [1], [1–3] or [1,3].
Why one reference „17. Zhao, J.; Tsuchikawa, S.; Ma, T.; Hu, G.; Chen, Y.; Wang, Z.; Chen, Q.; Gao, Z.; Chen, J. Modal Analysis and Experiment of a Lycium barbarum L. Shrub for Efficient Vibration Harvesting of Fruit. Agriculture 2021, 11, 519. https://doi.org/10.3390/agriculture11060519“?
The title of „Figure 1. 3D modeling of camellia oleifera tree“ is not correct. It has two parts: a and b. the same comment to „Figure 2. three-point bending test.“
All formulas must be cited by references. Or do they belong to the authors? (Equation 3, 9, 10 etc).
What units of strain measurement? (Figure 5).
The results of table 2 and figure 5 are not analyzed enough.
A broader analysis of the results is lacking.The text only says that such results were obtained. What is their meaning? (Section 3.4.3. correlation coefficient analysis etc)
The section on Discussion is very narrow. The results must be more analyzed and compared with the results of different authors. The one reference is used [17].
The conclusions are more general in nature. Novelty and importance should be highlighted.
Author Response
Author Response 2
Dear peer reviewers and editors
Hello! Thank you very much for your professional and wise comments on this article. According to the experts' questions and opinions, the author gives the detailed explanations in the form of one question and one answer, and marks the manuscript with red font in the corresponding position, as follows:
QUESTION1: Review one more time the spelling of letters in all text: title “Determination Of Vibration Picking” etc.
RE: Thank you very much for your valuable comments. The author has carefully checked and corrected the spelling of the words in the manuscript and marked it in red.
QUESTION2: In general, references are numbered in Arabic numerals in the texts. „...reference numbers should be placed ...before the punctuation; for example [1], [1–3] or [1,3].
RE: Thank you very much for your valuable comments. The authors have made changes to the format of the references in the manuscript and have numbered the references in Arabic numerals and marked it in red.
QUESTION3: Why one reference “17. Zhao, J.; Tsuchikawa, S.; Ma, T.; Hu, G.; Chen, Y.; Wang, Z.; Chen, Q.; Gao, Z.; Chen, J. Modal Analysis and Experiment of a Lycium barbarum L. Shrub for Efficient Vibration Harvesting of Fruit. Agriculture 2021, 11, 519. https://doi.org/10.3390/agriculture11060519”?
RE: Thank you very much for your valuable comments. When the author submitted the article, the references were numbered with Arabic numerals, possibly because there were some problems in converting the format.
QUESTION4: The title of „Figure 1. “3D modeling of camellia oleifera tree”is not correct. It has two parts: a and b. the same comment to “Figure 2. three-point bending test.”
RE: Thank you very much for your valuable comments. The author has changed the image title and marked them in red.
QUESTION5: All formulas must be cited by references. Or do they belong to the authors? (Equation 3, 9, 10 etc).
RE: Thank you very much for your valuable comments. Equation 3 is derived from the basic formula (i.e., the universal formula), and equations 9 and 10 are general expressions of the primary and quadratic equations, so the author does not need to quote them when writing the manuscript, because these formulas belong to the basic universal formula.
QUESTION6: What units of strain measurement? (Figure 5).
RE: Thank you very much for your valuable comments. Strain is a ratio, there are no units. It is the relative change in length.
QUESTION7: The results of table 2 and figure 5 are not analyzed enough.
RE: Thank you very much for your valuable comments. Table 2 shows the relevant physical characteristics of Camellia oleifolia. According to the test results obtained by the test method, the experimental process and steps have been clearly explained in the section of Materials and Methods. figure 5 shows the relationship between strain and branch diameter in the process of plastic deformation of camellia oleifera. The author has added related analysis content and marked them in red.
QUESTION8: A broader analysis of the results is lacking. The text only says that such results were obtained. What is their meaning? (Section 3.4.3. correlation coefficient analysis etc)
RE: Thank you very much for your valuable comments. The authors have added an analysis of the corresponding results and explained the implications of the results and marked them in red.
QUESTION9: The section on Discussion is very narrow. The results must be more analyzed and compared with the results of different authors. The one reference is used [17].
RE: Thank you very much for your valuable comments. The authors have added results for discussion with other articles and marked them in red.
QUESTION10: The conclusions are more general in nature. Novelty and importance should be highlighted.
RE: Thank you very much for your valuable comments. The author sorted out and summarized the conclusion, and listed the main contributions and highlights separately and marked them in red.
Thank you very much for your valuable opinions. Please review the revised article!
